# How Can We Improve the Vaccination Response in Older People? Part II: Targeting Immunosenescence of Adaptive Immunity Cells

**DOI:** 10.3390/ijms23179797

**Published:** 2022-08-29

**Authors:** Maider Garnica, Anna Aiello, Mattia Emanuela Ligotti, Giulia Accardi, Hugo Arasanz, Ana Bocanegra, Ester Blanco, Anna Calabrò, Luisa Chocarro, Miriam Echaide, Grazyna Kochan, Leticia Fernandez-Rubio, Pablo Ramos, Fanny Pojero, Nahid Zareian, Sergio Piñeiro-Hermida, Farzin Farzaneh, Giuseppina Candore, Calogero Caruso, David Escors

**Affiliations:** 1Oncoimmunology Group, Navarrabiomed, Instituto de Investigación Sanitaria de Navarra (IdiSNA), 31008 Pamplona, Spain; 2Laboratory of Immunopathology and Immunosenescence, Department of Biomedicine, Neurosciences and Advanced Technologies, University of Palermo, 90133 Palermo, Italy; 3Medical Oncology Department, Hospital Universitario de Navarra, Instituto de Investigación Sanitaria de Navarra (IdiSNA), 31008 Pamplona, Spain; 4Division of Gene Therapy and Regulation of Gene Expression, Centro de Investigación Médica Aplicada (CIMA), Instituto de Investigación Sanitaria de Navarra (IdiSNA), 31008 Pamplona, Spain; 5The Rayne Institute, School of Cancer and Pharmaceutical Sciences, King’s College London, London WC2R 2LS, UK

**Keywords:** adaptive immunity, immunosenescence, aging, vaccines, T cells, B cells

## Abstract

The number of people that are 65 years old or older has been increasing due to the improvement in medicine and public health. However, this trend is not accompanied by an increase in quality of life, and this population is vulnerable to most illnesses, especially to infectious diseases. Vaccination is the best strategy to prevent this fact, but older people present a less efficient response, as their immune system is weaker due mainly to a phenomenon known as immunosenescence. The adaptive immune system is constituted by two types of lymphocytes, T and B cells, and the function and fitness of these cell populations are affected during ageing. Here, we review the impact of ageing on T and B cells and discuss the approaches that have been described or proposed to modulate and reverse the decline of the ageing adaptive immune system.

## 1. Introduction

Ageing is one of the main health challenges worldwide, and promoting healthy ageing is a key global priority. The health of older people is threatened by their increased susceptibility to infectious disease and associated complications, which are related to many factors, especially the dysregulation of immunity generally termed “immunosenescence”. This is believed to adversely affect the efficacy of vaccines, thus reducing the protection provided by most current vaccines in older people. Specifically, a recent metanalysis indicated that the influenza vaccine’s effectiveness was 51% among people aged 18 and 64 years and 43% among people aged over 65 years [1]. Identifying the key factors responsible for reduced vaccination efficiency in older adults and devising countermeasures to solve this problem are essential for improving the outcomes of vaccination. This will allow for better protection against infections in this growing segment of the population [2].

To restore immunity in older people, several approaches have been assessed, including higher doses of antigens, new adjuvants and different routes of administration of antigens. Nevertheless, although stronger responses have been achieved by some of these means, the net result is still unsatisfactory [3,4]. Vaccine efficacy is usually based on antibody responses developed following immunization. However, cellular responses are also critical for long-term protection. Accordingly, T cells are the most important effectors in the cellular immune system, which are activated and mobilized by myeloid antigen-presenting cells (APCs) [5]. It is, therefore, of primary importance to design vaccination strategies specifically focused on older people. Developing countries recommend four vaccines for older individuals: influenza, pneumococcal infection, zoster and the combination against tetanus, diphtheria and pertussis [6]. All these formulations rely on a B cell response that is dependent on T cells, except for the *Streptococcus (S.) pneumoniae* infection, as it has a polysaccharide base [7].

In the present paper, we focus on the role of adaptive immunity in immunosenescence and on the possible strategies to revert immunosenescence, thus enhancing response to vaccines. The role of innate immunity in immunosenescence and the possible strategies to revert immunosenescence of innate immunity and thereby enhancing response to vaccines is treated by Aiello et al., 2022 [8].

## 2. Adaptive Immunity Immunosenescence

### 2.1. T Cells

CD3^+^ T cells arise in the BM and migrate to the thymus, where they mature and are selected before being exported to the periphery. The main function of T cells is to specifically recognize antigens, and this is mediated by T cell receptors (TCR). Broadly speaking, T cells can be classified into naïve T cells that respond to novel antigens, memory T cells that maintain long term immunity and regulatory cells, which regulate autoreactive responses. Naïve T cells express the lymph node homing receptor CCR7, while expression of C memory cells is divided based on the expression of protein tyrosine phosphatase receptor type C (CD45) isoforms in central memory (CD45RO^+^CD45RA^−^CCR7^+^) and effector memory (CD45RO^+^CD45RA^−^CCR7^−^) [9,10]. Naïve T cells migrate to lymphoid tissues, memory cells to secondary lymphoid organs and effector memory cells to the periphery [11]. First, antigens are presented to naïve T cells by professional APCs such as DCs. Then, these activated T cells produce interleukin 2 (IL-2), proliferate and differentiate into effector cells, which migrate to the sites of inflammation. Effector activated T lymphocytes have a short life span, but some of them survive and become memory T cells [12]. Of these, central memory T cells produce more IL-2 and are enriched in CD4^+^ T cells, while effector memory cells produce more effector cytokines and are enriched in CD8^+^ T cells. Central memory T cells are considered an intermediate state between naïve and effector memory cells [13].

CD4^+^ T cells (T Helpers) recognize peptides presented on MHC class II molecules by APCs. As a whole, they play a major role in instigating and shaping adaptive immune responses. Th1-polarised cells are responsible for control of intracellular pathogens such as viruses and some bacteria. Th2 polarized cells are important in the defence against large extracellular organisms such as helminths. Th17 cells are a recently discovered T helper cell subset, characterized by its production of IL-17, that have been linked to several inflammatory conditions [14]. CD8^+^ (cytotoxic) T cells recognize peptides presented by MHC Class I molecules, found on all nucleated cells. CD8^+^ T cells are involved in the immune defence against intracellular pathogens, including viruses and bacteria, and in cancer surveillance [15].

The accumulation of memory/effector cells observed with advancing age is determined by lifetime exposure to pathogens. It mainly concerns CD8^+^ T cells because, as discussed in the following paragraph, the main cause is persistent infection by HCMV, also responsible for the accumulation of terminally differentiated effector memory T cells (TEMRA). A group of T cells that express Fas cell surface death receptor (CD95) and interleukin 2 receptor subunit beta (CD122) have been described, which have characteristics of naïve cells but are in fact memory T cells. Of note, these stem-like T cells can differentiate into both types of memory cells [16]. T cells exhibit different roles in immunity along the life of a person. During infancy, the majority of T lymphocytes are naive and mature within the thymus. In this phase, T cells are selected and natural regulatory T cells differentiate to ensure systemic tolerance. Memory T cells accumulate and overtake naïve T cells until adulthood, and as their encounter with new antigens decreases, their production is stabilized. In this phase, the main role of lymphocytes is to maintain immune homeostasis.

Recently, the role of MAPK pathways in the functional competence of the immune system has been demonstrated, since these complex systems also regulate several functions of innate and adaptive immunity. They are also involved both in the production of pro-inflammatory cytokines and in the intracellular signalling cascades following the binding of cytokines to their receptors. Three main subgroups of MAPKs are known: mitogen-activated protein kinase (Erk), c-Jun NH2-terminal kinase (Jnk) and mitogen-activated protein kinase (p38). The p38-MAPK pathway stimulates the positive regulation of Th1 differentiation and polarization. This pathway is not active in Th2 cells [17]. Understanding the immune-regulatory functions exerted by MAPK pathways is critical to implementing integrative immunomodulatory strategies targeting these kinases (see Section 4). The mechanistic target of rapamycin (mTOR) also plays an important role in T cell activation and differentiation, especially of naïve CD4^+^ T cells in their differentiation toward T helper 1 cells (Th1) or Th17 phenotypes. The activation of the mTOR signalling pathway is under the control of TCR/CD28 (CD28 is one of the proteins expressed on T cells that provides co-stimulatory signals required for T cell activation and survival). The partial inhibition of mTOR could be beneficial for immune function in older people, although mTOR activity inhibits autophagy [18] (see Section 4).

### 2.2. Effect of Ageing and Viral Chronic Infections on T Cells

During the advanced years of life, some lymphocyte subsets become senescent with a loss of function, decreased proliferation capacity and an increase in chronic inflammations (Figure 1) [19]. Senescence features differ, in some aspects, from exhaustion. Senescent cells maintain cytotoxicity and can generate multiple cytokines, while exhausted cells have a limited function due to the constitutive expression of inhibitory receptors and a progressive loss of capacities for multicytokine production. The term “exhausted” cells is generally applied to terminally differentiated-senescent cells, with cell cycle arrest induced by the expression of the cell cycle regulator cyclin dependent kinase inhibitor 2A (p16INK4). However, these cells remain metabolically active, accumulate in multiple tissues during the ageing process and produce dysfunctional molecules [20,21]. Both types of cells arise from a continuous stimulation of the TCR [7,22]. During aging, the abundance of memory cells surpasses that of naïve T cells. Specifically, the thymus reduces its size and function (involution), a process that begins in infancy and reaches its maximum during puberty. Thymic regression includes changes in its architecture following its reduction in mass and number of thymocytes. Therefore, the number of naïve cells that arrive in the periphery is reduced and, in the meantime, memory cells expand clonally within the periphery. As a consequence, the diversity of the T cell repertoire decreases, as well as the capacity to respond to new pathogens [23]. Moreover, the process of negative selection that takes place in this organ is compromised, with an increase in self-reactive T cells, which supports autoimmunity and inflammaging [24]. Another cause of the shift in T cell populations is the life-long accumulated chronic antigen burden. In this case, the number of T cells that have not been exposed to an antigen decreases and the immunological space is filled by clonal expansion of memory and effector T cells through repetitive antigen exposure. Therefore, the repertoire that is available to confront new pathogens is diminished [25]. This is exemplified by HCMV, as this population represents the biggest proportion of memory cells in older people, especially for CD8^+^ T cells [26]. During the reactivation phase, HCMV causes clonal proliferation of HCMV-specific T cells, especially terminally differentiated CD8^+^ T cells, which typically lack CD28 expression. The presence of these cells results in impaired immune function, which adds to the effects of inflammation, in feedback. Thus, lifelong HCMV infection is the main factor responsible for peripheral expansion of Effector Memory and senescent/exhausted (TEM/TEMRA) T cells, but not for the age-related loss of naïve T cells [26,27,28]. The virus-specific effector memory cells, in particular, but not only, TEMRA, are poorly proliferative and highly cytotoxic, secreting cytokines. However, it is difficult to think that they are exhausted because they are able to perform their task very well, that of eliminating pathogens through cytotoxicity and cytokine secretion without proliferating [29].

Autophagy regulates various aspects of lymphocyte biology; thus, it is expected that age associated changes in this process too will affect T cells [30]. Autophagy is a mechanism to eliminate dead cells and debris, thereby this process is considered to counteract senescence. Macroautophagy and chaperon-mediated autophagy in T cells are diminished with aging in both CD4^+^ and CD8^+^ T cells, as well as in naïve and memory phenotypes. It should be emphasized that there are not enough studies to provide adequate detail for the molecular mechanisms underlying this phenomenon. T cells accumulate cellular damage and mitochondrial dysfunction as the individual ages, and these outcomes have been associated with impaired autophagy. Remarkably, autophagy deficient CD8^+^ T cells from mice fail to develop and maintain memory cell populations. Moreover, immune cells with impaired autophagy showed features of premature aging [31,32]. On the other hand, other mechanisms eliminate cellular and tissue debris through proteolytic enzymes and the proteasome. Accordingly, older people showed reduced expression of the proteases calpains and Tripeptidyl peptidase II (TPPII). Notably, TPPII-deficient CD8^+^ T cells were associated with premature immunosenescence. In addition, aged mice displayed impaired TCR signalling-related activation of the proteasome. Knockdown of selected proteasome subunits increased exhausted programmed cell death 1 (PD-1) positive CD4^+^ T lymphocytes [33,34].

Although immunosenescence is a common feature of the older population, there can be major individual differences. First, divergences are based on immunobiography, which comprises the immunological, clinical, socioeconomical and geographic history of each individual. Indeed, the individual genetic and epigenetic background interacts with different life-long immunological stimuli and, in this respect, HCMV seropositivity plays a relevant role as well as smoking status and cohabitation. Moreover, changes in the anthropological environment during the last century also condition immunological stimuli. The type, dose, intensity and moment of these stimuli will establish the ability of an individual to respond to a specific antigen [29,35,36]. Sex also impacts immunological aging. While oestrogens benefit the immune system, progesterone and androgen restrain it [37]. Accordingly, in terms of epigenetics, an appreciable number of immune response-related genes are localized on the X chromosome. Women have two copies of this chromosome, although one of them is partially silenced in each cell by a compensatory mechanism. Therefore, women possess more resistance against recessive mutations in these genes compared to males [38]. Although age-related epigenetic changes accumulate gradually, there are at least two periods in the adult lifespan during which the immune system undergoes more abrupt epigenomic changes. The first breakpoint manifests itself in both sexes similarly, whereas the second, latter breakpoint affects men more strongly and earlier than women [39]. Concerning immune cell populations, epigenetics favour T and B lymphocyte activity in women, while myeloid function is favoured in men. Immunosenescence features are more pronounced in men, including CD4/CD8 ratio, telomere shortening, proportion of HCMV-specific memory cells and inflammaging [40,41]. In general, there is an increased infection/mortality ratio in men vs. women in infectious diseases. On the other hand, some infectious diseases such as measles, toxoplasmosis, and dengue are more severe in women than in men (immunopathology effect). Female mice had better antibody responses to influenza, and studies on humans have established an immunosuppressive role for testosterone over responses to the influenza vaccine [42,43].

Aging affects both the composition of T cell populations and their phenotype (Figure 2 and Table 1). Naïve T cells lose their functional capacities and their capacity to regenerate, while highly differentiated memory T cells become the main population in charge of homeostasis in the periphery. The number of antigen-specific T cells increases with a loss of repertoire. Therefore, the response towards new antigens is compromised [44]. Virtual memory cells also have a higher representation, but their proliferation and function are impaired [45]. While CD8^+^ T cells increase their relative numbers, CD4^+^ T cells remain stable and are less affected by aging in terms of number of cells [46]. In addition, Th17 overtake Treg populations and favour a proinflammatory baseline state [47].

The expression profiles of TCRs are also significantly altered as the individual ages. Interestingly, the number of TCR molecules is altered in senescent T cells, but aged naïve T cells show diminished TCR diversity and altered TCR signalling [48]. Indeed, the composition of the cell membrane changes with a concomitant effect on TCR signalling. For example, the relative content of cholesterol in the membrane is higher, and therefore, TCR coalescence and recruitment of TCR-related proteins are compromised [49]. Accordingly, senescent CD4^+^ and CD8^+^ T cells show altered TCR-related extracellular signal-regulated kinases (ERKs) phosphorylation. Notably, aging increases the activity of the ERK negative regulator dual-specificity phosphatase 6 (DUSP6, phosphatases that can act upon tyrosine or serine/threonine residues) and the TCR is desensitized in CD4^+^ T cells [50]. It has been shown that DUSP4 expression generates defective responses by TCRs and increases the expression of senescence markers on T cells [51]. On the other hand, the inhibitory action of the protein tyrosine phosphatase non-receptor type 6 (PTPN6 or SHP-1), a signalling molecule, is reduced with aging, as well as LCK activity. Lck (lymphocyte-specific protein tyrosine kinase), a 56 kDa protein, is a member of the Src kinase family and is important for the activation of the TCR signalling in both naive T cells and effector T cells [52]. The MAPK p38 is constitutively active in aging T cells, and its activity is responsible for the repression of telomerase expression, proliferation and decreased expression of components of the TCR signalosome [53]. In a latter study, Lanna et al. generalised this mechanism by the formation of a macromolecular complex consisting of sestrins, a family of stress sensing proteins, bound to p38, ERK and JNK MAPK. The concerted action of these alternatively activated MAPKs caused the characteristic phenotypes of aged human T cells, which also included defects in DNA repair [54]. Indeed, sestrin 2 promotes the expression of the killer cell lectin-like receptor K1 (NKG2D)^−^ transmembrane immune signalling adaptor TYROBP (DAP12) complex on CD27^+^ CD28^+^ CD8^+^ T cells, which weakens TCR signalling and reprograms cells towards an innate-like phenotype [55]. TCR signalling is also compromised by inflammaging, and activated aged T cells show less calcium mobilization [56]. In addition, TNF-α production reduces the expression of the TCR/CD3 complex, phosphorylation of zeta chain of T cell receptor associated protein kinase 70 (ZAP-70), linker for activation of T cells (LAT) and phospholipase C gamma 1 (PLCγ), as well as reduced calcium mobilization. Of note, these features have been associated with ROS production [57]. On the other hand, in model studies, the depletion of Parkinsonism associated deglycase (PARK7/DJ-1), causing early-onset familial Parkinson’s disease, reduced oxidative phosphorylation and impaired TCR sensitivity in naïve CD8^+^ T cells at a young age, cumulatively leading to reduced evidence of aging in T-cell compartments [58]. The differential expression of TCRs is also altered. Sialomucin transmembrane molecule CD43 glycosylation is also affected, and this entails a hindrance for antigen detection [59]. Another molecule to consider is CD96, a type I membrane protein that may play a role in the adhesive interactions of activated T and NK cells during the late phase of the immune response. HIV patients show CD8^+^ T cells with senescent features when this receptor is downregulated [60]. Aged CD4^+^ T cells increase the expression of ectonucleoside triphosphate diphosphohydrolase 1 (CD39), and memory CD4^+^ T cells exhibit increased expression of CTLA-4 and PD-1 inhibitory receptors [7,61]. CD39 is an integral membrane protein that phosphohydrolyzes adenosine triphosphate (ATP) and, less efficiently, adenosine diphosphate (ADP), to yield adenosine monophosphate (AMP). This drives a shift from an ATP-driven proinflammatory environment to an anti-inflammatory milieu induced by adenosine [62].

Older individuals present memory T cells with elevated double strand DNA breaks and telomere shortening [53,54]. In fact, telomere length decreases as the cells become more differentiated [63]. Aged lymphocytes express the canonical senescence biomarkers p16INK4A and beta galactosidase, as well as proinflammatory molecules similar to SASP. Among these, aged T cells express increased levels of perforin, interferon gamma (IFN-γ), TNF-α, IL-1β, interleukin 18 (IL-18) and IL-6 [63,64,65,66]. The redox potential of aged T cells is also altered, and it has been associated with an increase in metallothioneins [57]. Furthermore, IL-2 production is reduced, as well as IL-2-related signal transducer and activator of transcription (STAT) phosphorylation. In contrast, as previously stated, the production of proinflammatory cytokines is augmented [67]. Apart from these features, the action of protein tyrosine kinases is altered, with proline rich transmembrane protein 2 (PKC), phosphatidylinositol 3-kinase (PI3K) and MAPK less activated with aging [53,68,69]. In addition, the complex C-terminal Src kinase (CSK)/peroxiredoxin 1 (PAG)/CD45 is deregulated and favours the inactive form of LCK [52]. Moreover, naïve T cells in older individuals express less HNF1 homeobox A (TCF1) through constitutive AKT-mTOR signalling, leading to reduced expression of miR-181 (see below), the acquisition of a senescent phenotype and the secretion of granzyme B exosomes [70]. In addition, the regulation of the lipid metabolism of naïve CD8^+^ T cells in older individuals is compromised [71]. Accordingly, senescent T cells show an unbalanced lipid metabolism that is mediated by an increase in phospholipase A2 group IVA (cPLA2) control by MAPK and signal transducer and activator of transcription STAT [72].

T cells also show epigenetic changes during aging. The differences in naïve and memory cell populations have been associated with an increase in histone acetylation and methylation [73]. Changes in DNA methylation in CD28^+^ CD4^+^ T cells have been associated with impaired function in aged cells, especially affecting cytokines and receptor expression [74]. Epigenetic signatures of aged T cells are different and might explain why CD8^+^ T cells are more affected by the process of senescence. Indeed, the signature of aged cytotoxic T cells is associated with less accessibility to basic cellular function genes [75]. Concerning miRNAs, older people express less miR-92A, and this correlates with the contraction of the naïve CD8 T cell compartment. A decline in miR-181 was associated with increased DUSP6 activity in aged CD4^+^ T cells (see Section 3), in addition to impaired TCR signalling and adaptive responses [50,76].

### 2.3. B Cells

B cells originate in the BM HSCs. In the CLPs and pre-pro B cells, rearrangement of the Ig heavy chain occurs, starting with D-J recombination. Then, in large pre-B cells, this is followed by V-DJ recombination, determining a functional heavy chain protein (Igµ) that associates with the surrogate light chains and the Igα/β dimer to express on cell membrane the pre-B cell receptor (pre-BCR). Intense proliferation and differentiation into the small pre-B cell stage is caused by signalling through the pre-BCR in large pre-B cells. V-J rearrangement of the Ig light chain then occurs in these small pre-B cells. That causes the production of a complete functional IgM BCR with a unique specificity. If immature B cells encounter antigens capable of cross-linking their BCRs, they are eliminated; this phenomenon is called Central Tolerance. Immature B cells IgM^+^ migrate to the periphery where they differentiate into long-lived mature follicular (FO) or marginal zone (MZ) B through distinct transitional stages. MZ B cells are innate-like B cells specialized to mount rapid T-independent, but also T-dependent, responses against blood-borne pathogens. They are also known to be the main producers of IgM antibodies in humans. Mature FO B cells recirculate between the secondary lymphoid where, following a cognate antigen encounter and interacting with CD4^+^ T cells in the germinal centre, they undergo rounds of proliferation accompanied by affinity maturation, resulting in a B cell pool which can bind to antigen with the highest affinity. The cells also undergo class-switch recombination to IgG, IgA and IgE. This switch diversifies B cell responses, matching the different immune challenges. Germinal centre B cells may differentiate into memory B cells or plasma cells. Most of these B cells will become plasmablasts (or “immature plasma cells”) and, eventually, plasma cells, and begin producing large volumes of antibodies. Memory B cells and plasma cells expressing somatically mutated and generally high affinity BCRs of switched isotypes exit the GC [77]. Mature B cells express different surface markers. In particular, it is possible to identify circulating naïve and memory B cells on the basis of the differential expression of IgD and CD27 surface markers, as follows: (1) IgD^+^CD27^−^ naïve B cells, (2) IgD^+^CD27^+^ memory unswitched B cells, (3) IgD^−^CD27^+^ memory switched (IgG^+^ or IgA^+^) B cells and (4) IgD^−^CD27^−^ double negative memory switched (IgG^+^ or IgA^+^) (DN) B cells, that show characteristics of senescent/exhausted memory B cells [20]. Serological memory is maintained by long lived plasma cells and by memory B cells being continuously re-stimulated by bystander cytokines producing-T cells as well as by microbial products triggering TLRs expressed on B cells, even in the absence of specific antigens [78].

### 2.4. Effect of Ageing and Viral Chronic Infections on B Cell

Age-associated changes are particularly evident in the BM of older people [79]. It has been widely demonstrated that the function of HSC significantly declines with age. In younger people, HSCs provide a balanced output of myeloid and lymphoid progenitor cells. The shift from lymphoid to myeloid differentiation that occurs with ageing determines a bias of older HSCs toward differentiation into common myeloid progenitor cells and a concomitant reduction in CLP frequencies; this is followed by a reduction in B and T cell production with ageing [80]. The reduction of naïve T lymphocytes is further linked to thymic involution (see Section 2.2). Moreover, inflammation has been shown to directly impair B lymphopoiesis by preventing B progenitors’ localization to the IL-7-enriched niches required for B cell development [81]. Thus, inflamm-ageing also contributes to decreasing the output of new B cells. Switched immunoglobulins (IgG and IgA) levels are positively age-related, whereas a decreased ability of older people and centenarians to produce IgD and IgM is reported, suggesting that in the older people, the B cell repertoire available to respond to new antigenic challenge is impaired [82]. B cell ageing is also characterized by an impaired ability to produce high affinity protective antibodies against infectious agents, due to both extrinsic and intrinsic defects. They are represented by impaired T-B cell signalling, or decreased somatic hypermutation and class switch recombination in germinal centre B cells [20,83]. These defects include the decreased expression and production of the E47 transcription factor, which controls the expression of AID, the activation-induced cytidine deaminase enzyme involved in somatic hypermutation and class switch recombination. The B cell repertoire age-related changes with consequent reduced diversity are responsible for expansion of oligoclonal populations and impaired response against foreign antigens with increased reactivity against autologous molecules, and, likely, reduced antibody affinity and specificity [84,85].

As in T cell compartment ageing, in the B cell compartment, an important role is also played by chronic viral infections by herpes viruses such as Epstein-Barr virus (EBV) and HCMV. EBV affects the B cell repertoire though a clonal expansion, whereas HCMV affects the levels of mutation of antibodies. The consequence is that the EBV or HCMV seropositivity and/or their reactivation are involved in the impairment of the humoral arm of the immune system in older people. HCMV-seropositivity, as well as inflamm-ageing, also cause the increase in both serum TNF-α and TNF-α-producing B cells, negatively correlated in vitro with the ability to respond to stimulation, measured by AID, and, in vivo, with serum response to the influenza vaccine [20,21,84,86].

“Core” subsets may be identified by evaluating only IgD and CD27 expression on CD19 B cells, as discussed above. Other authors, using other markers, have obtained different B subset results [87], making the need of using shared markers to uniquely characterize B cell subpopulations obvious [88]. IgD−CD27−DN B cells have also been reported to be expanded, in percentage, in other models of “chronic stimulation”, as in patients affected by some autoimmune diseases, in HIV-infected people and in Alzheimer’s disease patients. That strengthens the idea that the increase in percentage of the DN B cell population observed in older people should be related to inflamm-ageing. It can be stated that, as demonstrated for T cells (see Section 2.2), DN B cells are senescent/exhausted memory cells (however, see Section 2.2), and their expansion is the manifestation of a physiologic modification time-related (ageing) or a pathologic deregulation of the immune system [21,89].

With ageing, DN B cells express higher levels of CCR6 and CCR7 which are involved in the migration to the inflammatory sites, in this case, in chronic inflamed tissues. Thus, it is thought that the typical inflammatory milieu of older people influences the trafficking ability of B cells, rendering them more sensitive to both pro-inflammatory mediators over-produced in older people [21,90]. Therefore, in older people, IgD^−^CD27^−^ DN B cells are increased and show a trafficking phenotype that allows them to reach chronic inflamed tissues or tertiary lymph nodes. (Tertiary lymphoid organs are found at sites of chronic inflammation in autoimmune diseases such as systemic lupus erythematosus and rheumatoid arthritis. These organized accumulations of T and B cells resemble secondary lymphoid organs and generate autoreactive effector cells.)

The B cell compartment behaves as the T cell compartment (Table 1). It shows a shift in the magnitude of all B cell populations, a collapse in B cell receptor repertoire (BCR) diversity, correlated with a poor health status, a change in B cell dynamics and impairment of antibodies production, hence, a weakened humoral response. There is an age-related decrease in naïve B cells with an accumulation of B cells with the “exhausted/senescent” phenotype [21].

Antibody responses decrease with age, leading to increased frequency and severity of infectious diseases as well as to decreased protective effects of vaccines. Both high-affinity protective antibodies and the duration of protective immunity decrease with age. This decreased ability of older people likely results from combined defects in T cells, B cells and other immune cells, such as DCs [85].

With advancing age, the rate of “antigen-inexperienced” naïve T cell output dramatically declines due to thymic involution. Following their cognate antigen encounter, the differentiation of naïve T cells to an effector, highly differentiated and senescent phenotype is typically characterized by the loss of expression of CD28 and CD27, important co-stimulatory molecules for T-cell activation that bind to molecules expressed by APC. There is concomitant increase in terminally differentiated (TEMRA) cells, more evident in the cytotoxic compartment, due mainly to chronic viral infections, such as HCMV, that cause clonal expansion of these cells through repetitive stimulation. Other markers of senescence and/or exhaustion are represented by beta-1,3-glucuronyltransferase 1 (CD57), killer cell lectin-like receptor G1 (KLRG1) and check-point inhibitors PD-1 and leukocyte immunoglobulin-like receptor B1 (CD85j) (reciprocally, these cells lose CD28, CD27, CCR7 and IL-7Ra). Ageing has a strong impact on the remodelling of the B cell compartment, leading to impaired humoral immune responses. An evident effect is a significant reduction in the number of new B cells produced in the bone marrow, with a decrease in IgD^+^CD27^−^ naïve cells and the subsequent reduction in B cell repertoire diversity. The components of the memory B cell pool appear to be maintained with age, although some authors report otherwise, and these include IgM-only memory B cells, which are memory B cells that lose IgD expression, switched (IgD^−^/CD27^+^) and unswitched IgD^+^/IgM^+^ memory cells. Moreover, with ageing, an increase is observed in IgD^−^CD27^−^ double negative late memory B cells reported as exhausted memory cells that lose the expression of the CD27 memory marker. The different cell subsets are also described by various authors, also using other markers. Abbreviations: CM central memory; EM Effector Memory; EMRA, Effector Memory re-expressing CD45RA. References are in the text and in [21,29,89,90,91,92].

## 3. Immunosenescence of Adaptive Immunity and Vaccine Failure in Older People

Some of the features of immunosenescence have been associated with vaccine failure in older individuals. In general, changes in T cell function and sub-population shifts correlate with the antibody responses to the influenza vaccine [93]. In addition, antibody production is associated with lymphocyte infiltration but not with monocyte infiltration [94]. In the case of influenza, yellow fever and hepatitis B, there is a lower response when the baseline state of inflammation is high [95]. Furthermore, individuals with decreased CD28 expression in T cells show diminished T cell proliferation and weaker responses after vaccination. This is coupled with decreased TCR repertoire diversity, which indicates increased susceptibility to new pathogens and the reduced efficacy of vaccination programs [96]. Additionally, diminished CD4/CD8 ratios correlate with poor antibody titres against influenza. The effect of the presence of HCMV-specific T cells or HCMV infection in older individuals is unclear. While some studies indicate that HCMV positive older individuals present worse vaccine responses, this is not the case in younger individuals. Other studies report better vaccine responses in the seropositive individuals [97,98]. On the other hand, a single-nucleotide polymorphism has been found to be associated with reduced transcriptional transactivation of CD39, which, in turn, correlates with better responses to influenza and varicella zoster virus immunizations [61].

Effective vaccines need to elicit good T cell responses. Memory T cells have a broad specificity against internal and conserved pathogen epitopes due to their longevity and residence in circulation and peripheral sites [99]. For example, for tuberculosis and smallpox vaccines, T cell responses are essential [100,101]. In addition, BCG vaccines were reported to generate adequate CD4 and CD8 T cell responses [102]. In fact, most immunosenescence research has been focused on T cell-mediated immunity [103]. DNA vaccines in particular generate good T cell responses, as the antigen is expressed from a gene inside the target cells, leading to a strong T cell response, which in turn culminates in good antibody responses. However, outcomes from clinical trials were not encouraging [104]. Viral vectors contain a genetic load that is introduced into the target cells at the vaccination sites, eliciting strong transgene expression leading to adequate T cell responses [105,106].

The efficacy of influenza immunization in the adult population is about 59%, while this percentage decreases to 39% in individuals over 65 years old [107]. Immunity against influenza includes CD8^+^ T lymphocytes that recognize internal viral antigens as well as antibodies towards surface proteins such as hemagglutinin. In addition, the presence of effector-memory and effector T cells in the respiratory tract is important for protection against re-infections [108]. While the expression of genes associated with T and B lymphocyte function correlates positively with antibody responses, the genes associated with inflammation and monocytic-lineage have a negative impact [109]. It should be noted that aged CD8^+^ T cells demonstrated decreased proliferative capacity, and this reduction in influenza-specific CD8^+^ T cells negatively affects viral clearance in older patients [110]. Moreover, older individuals show an impaired CD4 response after vaccination [111]. The number of Th1 cells that secrete inflammatory cytokines is reduced in the lungs of aged mice [112]. Furthermore, CD8 TCR diversity in aged mice is lower, which is reflected in a reduced response against epitopes from the nucleoprotein of the influenza virus [113]. Regarding pneumococcal immunization, the efficacy is around 40–65% for older people, contrasted to the 60–70% efficacy for younger individuals [114]. The pneumococcal vaccine is a polysaccharide-based antigen formulation, which is independent of T cells. However, T cells are important in the natural immunity against these bacteria. Accordingly, CD4^+^ T cells secreting IL-17 mediate the adaptive response against *S. pneumoniae* [115]. On the other hand, varicella zoster virus reactivates throughout life, but individuals are asymptomatic due to T cell-dependent cellular immunity [116]. In natural infections, antigen-specific CD4^+^ T cells are generated, and clinical trials show that the CD4 response persists for three years after vaccination [117]. Specifically, a vaccine with the adjuvant AS01B has been developed that favours T cell responses in animal models [118]. Notably, virus attenuated vaccines have been shown to generate an adequate TCR repertoire in CD4^+^ T cells [119].

Although most studies attempting to explain the role of immunosenescence in vaccine response have centred on immunosenescence T, immunosenescence B also plays a role. Humoral immunity is known to play an important role in preventing influenza virus transmission and infection, and immunogenicity of influenza vaccines is usually measured by hemagglutinin (HA) inhibition (HAI) assay, which quantifies antibodies specific for the virus HA. A greater number of older adults fail to seroconvert, i.e., to have the four-fold increase in post-vaccination antibody titre, relative to their younger counterparts, which is one of the WHO criteria for assigning responsiveness, with seroconversion rates ranging from 10 to 30% in older adults compared to 50–75% in younger subjects (although it can sometimes be the case that older people already have a high antibody titre due to previous exposures, and thus cannot increase titres four-fold, resulting in erroneous classification as non-responders). In particular, older adults may fail to generate protective HAI antibody titres compared to younger adults. Cellular immunity is also strongly associated with protection against influenza, as some older adults have been shown to remain protected against infection even in the absence of robust antibody responses. However, further studies are needed in order to fully understand the effects of immunosenescence on cellular immunity to influenza [108,120,121].

As reported by Frasca and Bloomberg [86], several studies have identified B cells’ intrinsic defects that account for sub-optimal antibody responses of older people. They are: (i) the decrease in class switch recombination, responsible for the generation of a secondary response; (ii) the decrease in de novo somatic hypermutation of the antibody variable region, responsible for the failure of high affinity antibodies; (iii) decreased binding and neutralization ability, as well as binding specificity, of the secreted antibodies; (iv) increased epigenetic changes associated with lower antibody responses and (v) increased frequencies of inflammatory B cell subsets.

Human studies have found that people > 65 years old have significantly lower antibody titres against many of the common pneumococcal serotypes and diminished opsonisation activity compared to younger adults. Therefore, antibody titres wane over time, and there may also be functional deficiencies in antibody responses against pneumococcal antigens. While humoral immunity is primarily thought to mediate protection from disease, there are conflicting reports regarding age-related changes in T cell responses against pneumococcal infection [121].

It was demonstrated that inflamm-ageing plays an important role in compromising the immune responses by way of inducing the high expression of some microRNAs that interfere with B cell activation. In vitro, this drives TNF production and inhibits B cell activation. Increased serum levels of TNF are also linked to a defective T cell response, in part due to reduced expression of CD28 on T cells. Moreover, in monocytes, the pre-vaccination expression of genes related to inflammation and innate immune response is negatively correlated to vaccination-induced activation of influenza-specific antibody responses [94,122,123].

Nowadays, we are immersed in a pandemic caused by the SARS-CoV-2 virus, and older people constitute one of the main risk groups affected (74.3% of deaths in the US) [124]. Remarkably, some species from the coronavirus family induce thymic involution, and thus the hypothesis that this might also occur with SARS-CoV-2 virus has been considered [125]. In addition, T cell response is critical for immune protection against SARS-CoV-2, since it is essential for viral clearance, prevention of infection and recognition of viral variants [126]. In this regard, severe COVID-19 disease has been associated with lower TCR diversity against SARS-CoV-2 epitopes, accompanied by reduced T cell responses [110]. Specifically, an increase in TEMRA CD8^+^ T cells (see Section 2.1) is associated with worse memory responses against SARS-CoV-2, as well as a compromised antibody response [127]. In general terms, antibody responses are decreased in older individuals who are prone to mild and moderate adverse events [128]. It is noteworthy that SARS-CoV-2 is demonstrated to reduce the number of CD8^+^ T lymphocytes, a fact that is associated with poor survival of COVID-19 patients [129]. Accordingly, impaired cytotoxic CD8 T cell responses have been reported in older COVID-19 patients [130]. Although SARS-CoV-2-cross reactive CD8^+^ T cells have been detected in unexposed individuals, this population was found to be decreased in older individuals [131]. SARS-CoV-2 was also reported to decrease CD4^+^ helper T cells, specifically in older patients suffering severe COVID-19 disease [132]. Vaccines have been developed and approved to prevent the severity of the infection, but, unfortunately, an aged immune system compromises their efficacy. For mRNA-based vaccines, the BNT162b2 formulation first claimed that 18–98 years old individuals benefited with a 94% efficacy within the >65 years old group [133,134]. However, a later study demonstrated the time-dependent lessening of both cellular and antibody responses in older people compared with younger adults and a relationship with inflammaging [135]. The second dose of the BNT162b2 mRNA vaccine showed no differences in neutralization potency against the B.1.1.7 (Alpha), B.1.351 (Beta) and P.1. (Gamma) variants of concern in comparison with the wild type virus in older individuals. However, SARS-CoV2-spike specific T cells from older participants produced less IFN-γ and IL-2 [136]. Alternatively, the mRNA-1273 vaccine generates adequate antibody titres independently of the age of the vaccines [137]. Finally, the adenovirus-based vaccine AZD1222 has been demonstrated to be a safe and tolerated alternative, with a degree of immunogenicity comparable to younger individuals [138].

## 4. Strategies to Reverse Immunosenescence of Adaptive Immunity in Older People

A significant effort has been made to modulate T cell senescence by a wide variety of strategies (Table 2). First, a group of approaches target altered molecules in the senescent T cells. On TCR signalling, genetic inhibition of DUSP6 or DUSP4 recovers T cell signalling in aged T lymphocytes [139]. CD4 responses can also be improved by DUSP6 kinase repression by its natural inhibitor miR-181a, or specific siRNA [50]. On the other hand, pharmacologic inhibition of SHP-1 allows increased secretion of IL-2 and proliferation of CD4^+^ T cells [52]. The function of the TCR can also be recovered by targeting the p38 MAPK pathway. The MAPK p38 blockade reverses CD8 senescence by a process independent of mTOR [140]. Moreover, the simultaneous inhibition of this MAP kinase and PD-1 favours the proliferation of effector-memory CD8^+^ T cells that re-express CD45RA (TEMRA) [141]. In addition, the proliferation of highly-differentiated human T cells is recovered by inhibiting the macromolecular complex made of protein kinase AMP-activated (AMPK), TGF-beta activated kinase 1 (MAP3K7) binding protein 1 (TAB1) and p38, either with AMPK or with p38 inhibitors [53]. In aged mice, T cell activity is recovered upon knockout of the sestrin-MAPK activation complex (sMAC), thus increasing the efficacy of the influenza vaccine FLUAD. In addition, TCR signalling is restored by genetic blockade of sestrins [54,55]. MAPK p38 inhibitors have been used to reduce the inflammation generated by the antigens from a purified derivative of tuberculin, Candida albicans and Varicella virus [142]. Another strategy that restores TCR sensitivity and prevents immuno-aging is DJ-1 inhibition at a young age [58]. Other membrane receptors are compromised in aged T cells, and thus, their modulation has also been evaluated. Accordingly, PD-1 suppression increases cytokine production [7]. Furthermore, the inhibition of tumour necrosis TNF-α or its receptor postpones the CD28 down-regulation characteristic of T cell replicative senescence [143].

Targeting cell signalling is an interesting strategy to reverse T cell immunosenescence. Thus, mTOR inhibition improves general immune response after vaccination of older people. Accordingly, clinical trials using the mTOR inhibitor everolimus showed a better immune function and lower frequencies of PD-1 T lymphocytes after immunization against influenza [144]. Moreover, the combination with PI3K inhibitors increased the control of infection in older people. Inhibition of the mTOR upstream activator VPS39, a protein that may promote clustering and fusion of late endosomes and lysosomes, ameliorates the expansion of antigen-specific T cells, generating higher levels of memory T cells [145]. mTOR inhibitors have also been tested in combination with SARS-CoV-2 vaccines in immunocompromised individuals. The combination treatment showed better antibody and T cell responses compared to the absence of mTOR inhibition [146].

Another approach relies on the use of the autophagy activator spermidine, which favours the expansion and functions of antigen specific CD8^+^ T cells in aged mice [89].

It has been demonstrated that the endogenous polyamine metabolite, spermidine, induces autophagy in vivo, hence, rejuvenating memory B cell responses [147]. Data from mice and humans indicate that spermidine has the potential to be safe for testing its epigenetic-dependent and independent effects on human health span. Spermidine post-translationally modifies the translation factor eIF5A, essential for the synthesis of the autophagy transcription factor TFEB that coordinates expression of lysosomal hydrolases, membrane proteins and genes involved in autophagy. Spermidine is depleted in older people and its supplementation restored TFEB expression and autophagy, hence, improving the responses of B cells from older people. Taken together, these results reveal an unexpected autophagy regulatory mechanism at the translational level, which can be used to block and/or reverse human immunosenescence.

Furthermore, the restoration of lipid metabolism by blockade of cPLA2, or the use of drugs that favour lipid catabolism, prevent T cell decline [71,72]. The blockade of TNF-α, or its receptor, postpones CD28 down-regulation in replicative senescent T cells [143]. Alternatively, exosomes derived from placenta mesenchymal stem cells (MSC) containing miR-21 induce loss of expression of senescence markers in CD4^+^ aged T cells by activating the phosphatase and tensin homolog (PTEN)/PI3K-NFE2-like bZIP transcription factor 2 (Nrf2) signalling axis [148]. The AMPK agonist metformin could be considered an interesting treatment to reverse T cell immunosenescence, as it reduces inflammation by decreasing Th17 differentiation and increasing Tregs [149].

A second group of approaches does not directly target senescent T lymphocytes. However, some of these methods accomplish suitable T cell functions in vaccination, as mentioned above. For example, the combination of lipophilic adjuvants and toll-like receptor 4 (TLR4) agonists improves T follicular responses to malaria vaccines in mice [150]. The addition of the adjuvant AS01 in the vaccine formulation for herpes zoster virus increases the number of CD4^+^ T cells in older individuals [151]. A systematic review from 2022 suggests that adjuvanted influenza vaccines are preferable over conventional vaccines for older individuals [152]. MF59 and AS03 adjuvants have been used for influenza vaccines for older adults with great effectiveness [153]. Influenza vaccines that contain MF59 adjuvant showed better persistence of B cell and CD4^+^ T cell responses, and similar effects have been described for AS03 [154,155]. Flagellin from *Salmonella typhimurium* has also shown promising results in influenza vaccine formulations with an increased number of IFN-y producing memory CD4^+^ T cells [156]. AS02, GLA-SE and CpG are TLR4 and TLR9 agonists that generate better humoral immunity in pneumococcal vaccines [157]. GLA-SE induces Th1-biased T cell responses and enhances cytokine and granzyme B secretion [154]. On the other hand, Imiquimod, a TLR7/8 agonist that improves influenza vaccines’ protection, was shown to increase IFN-y expression and IgG isotype switching [158].

Interestingly, immune fitness can be modulated by lifestyle. Thus, exercise is associated with fewer senescent lymphocytes, better function of these cells and improved response after vaccination [159,160]. With respect to the effects of exercise effects, one potential may be increase in thymic mass and the resulting output of naïve T-cells in older people through increased levels of IL-7 and/or growth hormone synthesis. IL-15 released from muscle, now considered an important regulating organ for the immune system, can improve NK cell cytotoxicity and cytokine secretion, helping to maintain blood T and NK cell numbers. Skeletal muscle produces myokines, proteins with anti-inflammatory and immune response enhancing effects [29]. Moreover, apoptosis of exhausted T cells could be caused by frequent bouts of exercise. Notably, the consumption of specific nutrients is also beneficial. Accordingly, zinc deficiency generates a decrease in timulin, a peptidic hormone that enhances the expression of T cell activation markers. Indeed, zinc supplementation reduces infection incidence and increases CD4 and CD8 numbers in older individuals [161]. Vitamin E intake is associated with IL-2 production and naïve T cell activation and proliferation [162]. Moreover, vitamin C might counteract inflammaging and help T cell maturation [163]. On the other hand, carotenoid supplementation in older people allows T cells to express a mature phenotype. Thus, high doses of β-carotene increase CD4^+^ T cells [164]. Polyphenols have been described to increase IL-2 and IFN-gamma and correlate with improved immune responses. Accordingly, CD4^+^ T cell numbers are increased in aged rats after resveratrol dietary intake [165]. The consumption of polyunsaturated fatty acids induces the proliferation of T lymphocytes [89].

Adoptive T cell therapy has also been considered for reversal of immunosenescence. Stem cell memory cells or virus-specific T cells might be expanded ex vivo and transferred to older people [166]. Another adoptive cell therapy is based on mesenchymal stem cell (MSCs). MSCs are multipotent progenitor cells with the ability to reduce the inflammation. In addition, these cells express transforming growth factor beta 1 (TGF-beta), a molecule that promotes the generation of CD8 and Tregs. This approach has been effective in disease models associated with an impaired effector T cell response or immune regulation mediated by Tregs [167]. For example, exosomes derived from placental MSCs reduce the expression of senescence markers in aged CD4^+^ T cells. This outcome relied on the miR-21, a microRNA that activates the PTEN/PI3K-Nrf2 axis, a signalling pathway that has been implicated as an important regulator of the proliferation, differentiation and apoptosis in a variety of cell types [148]. Thymic negative selection can be restored by the transplantation of these extracellular vesicles from young mice to older animals, leading to moderation of inflammaging [168]. In addition, older COVID-19 patients treated with thymosin alpha 1, a polypeptide hormone secreted by epithelial thymic cells, experienced an increase in CD4^+^ and CD8^+^ T cells [169]. On the other hand, the so-called “anti-aging drugs”, senolytics, promote removal of senescent immune cells that accumulate with age [170]. Specifically, in old mice, it has been demonstrated that senolytic drugs can impact CD4^+^ T cells, most likely by modulating the microenvironment, which can then positively influence T cell differentiation during the response to influenza infection [171].

Finally, another study proposed a different strategy to enhance immune responsiveness in aged mice and older humans, through rejuvenation of the B lineage upon B-cell depletion. The authors used old and young mice to deplete blood B cells, analysed B cell subgroups, their repertoire and cell functions in vitro and immune response in vivo. Depletion of B cells in aged mice resulted in a rejuvenated B cell population generated de novo in the bone marrow. Rejuvenated B cells exhibited a “youthful” repertoire and cellular reactivity to immune stimuli in vitro. However, the treated mice did not increase antibody responses to immunization in vivo, nor did they survive longer than the control mice in a “dirty” environment. Older patients, previously treated with rituximab, healthy older and younger subjects were vaccinated against hepatitis B (HBV) after undergoing a detailed analysis for B-cell compartments. Consistent with the results obtained in models, B cells from older depleted patients showed a “young”-like repertoire, population dynamics and cellular responsiveness to stimulus. However, the response rate to HBV vaccination was similar between depleted and nondepleted patients, although antibody titres were higher in depleted patients [172]. Further studies are necessary to apply this approach for enhancing humoral immune responsiveness of older people.

## 5. Conclusions

In the present review, we show the effects of immunosenescence on adaptive immunity, the consequences that this phenomenon has on vaccines, as well as the strategies that target this process to enhance immune response to vaccines in older people. The detrimental consequences of ageing on the adaptive immune system begin with the insufficient generation of T and B lymphocytes from hematopoietic stem cells (HSCs). In addition, the chronic antigen exposure of both T and B cells during the whole life of the individual also determines the quality of this response. On the other hand, the cellular mechanism controlled by the signalling pathways, membrane receptors and epigenetics is also compromised due to ageing in these cell populations. It is important to emphasize that T and B lymphocytes collaborate and are not independent. The result is the inadequate adaptive immune response observed in older people and the failure of vaccination strategies in this segment of the population. However, as these features have been related to the lower fitness of vaccines in older people, several promising approaches have been developed. Specifically, we have described several molecular approaches that target this impaired signalling. Another promising approach is the replacement of aged cells by new ones with better fitness, similar to the adoptive cell therapies used in cancer treatment. Moreover, the depletion of impaired cell populations has also been discussed. It should be noted that even though hallmarks of senescence of adaptive immunity are well known, they have similarities with the process of exhaustion. Therefore, a further analysis of the differences between senescence and exhaustion is required. In this regard, the ISOLDA consortium was formed in January 2020 in order to combine forces to improve vaccination strategies for older people. ISOLDA comes from the acronym “Improved Vaccination Strategies for OLDer Adults”. It is a consortium of seven partners on a joint mission to develop better-tailored vaccines against viral diseases in adults aged 65 years or older. The aim of the project is to lower the morbidity and mortality burdens in this age category.

## Figures and Tables

**Figure 1 ijms-23-09797-f001:**
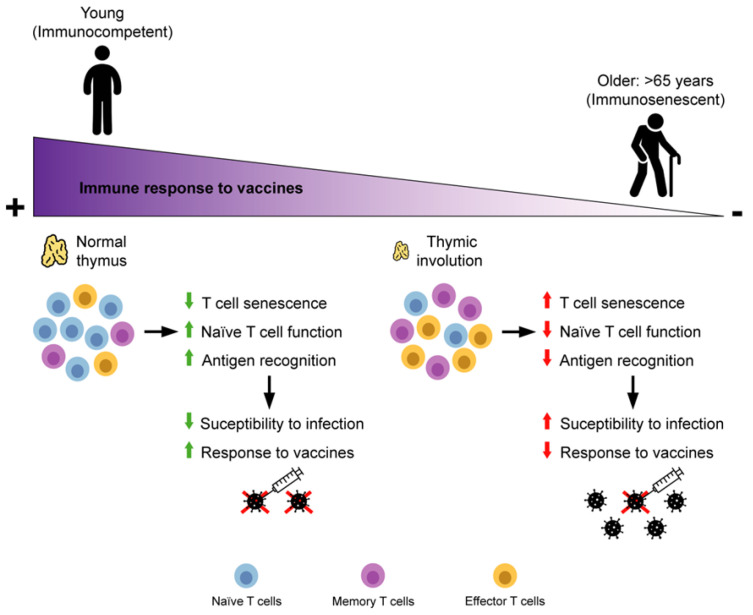
T cell immunosenescence and response to vaccines. The figure summarizes the major characteristics of immunological aging. The degree of response to vaccination over time is schematically represented on top. Below, the anatomical and cellular characteristics associated with aging are represented.

**Figure 2 ijms-23-09797-f002:**
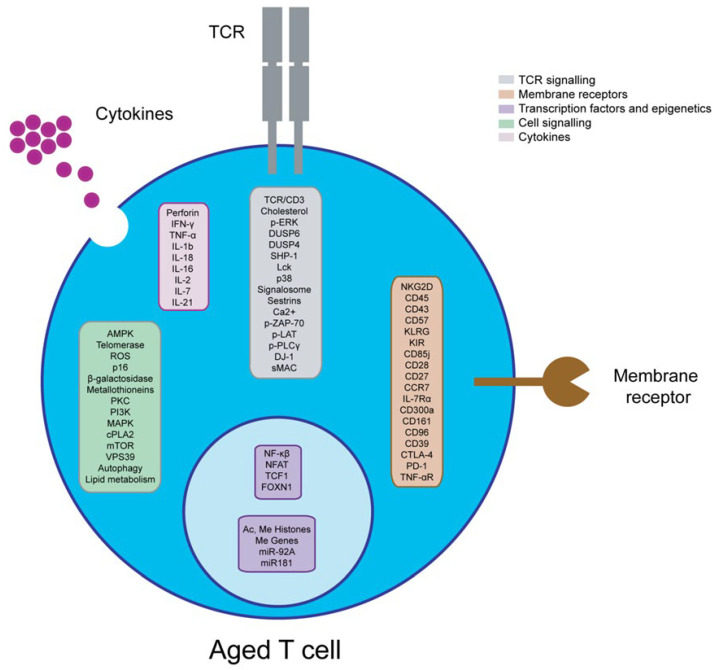
Schematic representation of altered molecules in senescent T cells. During T cell senescence, multiple molecules are altered, among them molecules related to TCR signalling (grey), cell signalling (green), epigenetics and transcription factors (purple), membrane receptors (brown) and cytokines (pink).

**Table 1 ijms-23-09797-t001:** Age-associated changes in adaptive compartments.

Cell Phenotype	Changes	Causes and/or Effects
**T cell** CD3+	=/↓	Reduction of haematopoietic stem cell progenitors; Defects in thymic stromal niches. Reduced T-cell responses;
**CD4^+^ Naïve** CD4^+^/CD45RA^+^/CCR7^+^/CD27^+^/CD28^+^	↓	Thymic involution; Phenotypic conversion of naïve T cells into memory phenotype. Reduced responses to new antigens and neoantigens; Increased susceptibility to infections.
**CD8^+^ Naïve** CD8^+^/CD45RA^+^/CCR7^+^/CD27^+^/CD28^+^	↓↓	
**CD4^+^ T_CM_** CD4^+^/CD45RA^−^/CCR7^+^/CD27^+^/CD28^+^	=/↑	Effects of immunobiography. Reduced responses to cognate antigens; Increased susceptibility to infections, autoimmune disorders, chronic diseases, cardiovascular disease and cancer.
**CD8^+^ T_CM_** CD8^+^/CD45RA^−^/CCR7^+^/CD27^+^/CD28^+^	=/↑	
**CD4^+^ T_EM_** CD4^+^/CD45RA^−^/CCR7^−^/CD27^−^/CD28^−^	=/↑	
**CD8^+^ T_EM_** CD8^+^/CD45RA^−^/CCR7^−^/CD27^−^/CD28^−^	↑↑	
**CD4^+^ T_EMRA_** CD4^+^/CD45RA^+^/CCR7^−^/CD27^−^/CD28^−^	=/↑	Reactivation of persistent virus infections. Reduced responses to cognate antigens; Increased susceptibility to infections, autoimmune disorders, chronic diseases, cardiovascular disease, and cancer.
**CD8^+^ T_EMRA_** CD8^+^/CD45RA^+^/CCR7^−^/CD27^−^/CD28^−^	↑↑	
**B cells** CD19+	↓	Reduction in haematopoietic stem cell progenitors; Reduced B-cell responses.
**Naïve** CD19+/IgD^High^/IgM^High^/CD27^−^ or CD19^+^/IgG^−^/IgA^−^/CD27^−^	↓	Phenotypic conversion of naïve B cells into memory phenotype; Increased susceptibility to infectious diseases; Reduced ability to respond to new pathogens and reduced protection of vaccination.
**Memory unswitched** CD19+/IgD^Low^/IgM^High^/CD27^+^	=	Maintained immune response against well-known antigens.
**Memory switched** CD19+/IgD^−^(Switched Igs, IgG^+^/IgA^+^/IgE^+^)/CD27^+^	=/↓	
**IgM-only memory** CD19+/IgD^−^/IgM^+^/CD27^+^	=/↓	
**Double negative** CD19+/IgD^−^/(Switched Igs, IgG^+^/IgA^+^/IgE^+^)/CD27^−^	↑	Negatively associated with the serum response to the influenza vaccine; Secretion of pro-inflammatory cytokines

**Table 2 ijms-23-09797-t002:** Strategies to reverse immunosenescence of adaptive immunity in older people.

Strategy		Effect
**Senescent T cell**	DUSP6 inhibition	Recovery of T cell signalling
	DUSP4 inhibition	Recovery of T cell signalling
	SHP-1 inhibition	Increased secretion of IL-2 and proliferation of CD4^+^ T cells
	MAPK p38 inhibition	Reversion of CD8^+^ T cell senescence
	MAPK p38 and PD-1 inhibition	Proliferation of TEMRA CD8^+^ T cells
	AMPK-TAB1-MAPK p38 complex inhibition	Proliferation of highly-differentiated T cells
	Sestrins–MAPK complex inhibition	Recovery of T cell activity Increase in influenza vaccine efficacy in mice
	Sestrins inhibition	Recovery of TCR signalling
	DJ-1 inhibition	Restoration of TCR
	PD-1 inhibition	Increase of cytokine production
	TNF-alpha inhibition	Postponement of CD28 downregulation
	mTOR inhibition	Improvement in immune response after influenza and SARS-CoV 2 vaccination
	mTOR and PI3K inhibition	Control of infection
	VPS39 inhibition	Higher levels of memory T cells
	Autophagy inhibition	Expansion of antigen specific CD8^+^ t cells
	cPLA2 inhibition	Prevention of T cell decline
	PTEN/PI3K-NRF2 axis activation	Loss of senescence markers expression
	AMPK activation	Decrease of Th17 differentiation and increase in Tregs
	Senolytic drugs	Depletion of senescent cells
	Thymosin	Increase in CD4^+^ and CD8^+^ T cells in older COVID-19 patients
**Adjuvants**	Lipophilic adjuvants and TLR4 agonist	Improvement of T follicular responses to malaria vaccines in mice
	AS01 adjuvant	Increase in CD4^+^ T cells for herpes zoster virus vaccination
	MF59	Persistence of B cell and CD4+ T cell responses
	AS03	Persistence of B cell and CD4+ T cell responses
	Flagellin	Increase in IFN-γ producing memory CD4+ T cells
	GLA-SE	Th1-biased T cell responses and enhances cytokine and granzyme B secretion
	Imiquimod	Increase in IFN-y expression and IgG isotype switching
**Lifestyle**	Exercise	Decrease in the number of senescent lymphocytesIncreased levels of IL-7 and IL-15 Apoptosis of exhausted T cells
	Zinc	Increase in CD4 and CD8 numbers
	Vitamin E	IL-2 production Naïve T cell activation and proliferation
	Vitamin C	Reduction of inflammaging T helper maturation
	Carotenoid	Mature T cell phenotype
	Polyphenols	Increase in IL-2 and IFN-gamma
	Polyunsaturated fatty acids	Proliferation of T lymphocytes
**Adoptive T cell therapy**	Stem cell memory cells	
	Virus-specific T cells	
	Mesenchymal stem cells	Reduction in the expression of senescent markers in CD4^+^ T cells Moderation of inflammaging in mice
**B cell**	Induction of autophagy	Improvement in B cell response
	Depletion	Rejuvenation of B cell population

References in the text.

## Data Availability

Not applicable.

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
