# Peer review of "How Can We Improve the Vaccination Response in Older People? Part II: Targeting Immunosenescence of Adaptive Immunity Cells"

_ijms, 2022, doi:10.3390/ijms23179797_

Round 1

Reviewer 1 Report

General comments

The manuscript submitted by Garnica et al., is a review contribution with the attempt to improve the efficacy of the vaccines administered to older people, by targeting immune-senescence of adaptative immunity. The manuscript is a highly comprehensive compendium including a description of adaptative immunity cell types and signaling pathways involved in its regulation, such as MAPK p38, ERK and NK. The review addresses the T cell receptors, cytokines, interleukins, cell receptors, and mediators in aged T cells. Also, the B cell development branch is extensively described. The second part is focused on the immune-senescence of adaptative immunity and vaccine failure in older people, and potential strategies to reverse the adaptative immune-senescence in them. The review is excellent, highly educative and full of novel ideas to approach the immune-senescence problem. Furthermore, this is an important area of research, as the elder people numbers are rapidly increasing in all countries. This revision could be very useful to both beginners in the area and also expert ones.

I missed aditional information in two specific areas, one dealing with the use of specific adjuvants that can mitigate the problem, and a second one specifically addressing the problem of SARS-CoV-2 and related deadly beta-coronaviruses. It is clear that reference to both subjects is made during the review, but due to their relevance both in their efficacy in the application (adjuvants) and because the relevance of the problem these days (CoVs), the incorporation of recent data would enhance the present quality of the review.

Specific points

1.     Lines 376-402. Perhaps areas that are highly speculative as the one included within the indicate lines could be limited in the text.

2.     Line 496 S. pneumoniae should be in italics.

3.     Line 497. It seems that this sentence is incomplete.

4.     Line 505. Please, note that humoral immunity not only plays a role in the protection against the majority of virus infections, but that in general it takes care of the elimination of more than 85 % of infectious viruses.

5.     Along the text there are many minor misspellings, this is the case in lines: 413, 436, 462, 463, 464, 465, 687, 497, 522, 623, and 687.

Reviewer 2 Report

This work jointly co-first authored by Garnica and Aiello and their colleagues provides a very comprehensive review on adaptive immunity related to vaccination response, with a particular emphasis on the effect of ageing. This review is going to be an important reference to the field. I have several minor suggestions for the authors to consider to better their work-

1.       Title: The phrase “II part” looks weird and does not read coherently with the rest of the title. In addition, considering that a substantial proportion of this work is on basic immunology, the title could be enhanced to reflect this point.

2.       Introduction, first paragraph: the authors stated that vaccine effectiveness is lower in older adults. While this is generally true, a more objective deliberation would be desired, for example, by citing figures of influenza vaccine effectiveness in different age groups.

3.       Introduction: It is stated that “The evaluation of vaccine efficacy is usually based on antibody responses developed following immunization.”. I am afraid this is not true as vaccine efficacy is determined by outcome (infected or not) in clinical trials instead of by laboratory parameters such as antibody levels.

4.       Consistent terminology is essential to avoid misunderstanding. For example

     a.       CD4 T cells should have read as CD4+ T cells;

     b.       TMERA should have read as “Terminally differentiated effector memory T cells”, not “CD8+ terminally differentiated”.

     c.       CD27+ CD28+ CD8 (or CD8+ or subset of CD8+?) T cells  

     d.       Just name a few. Please proofread.

5.       Section 2.4: “the Activation-induced cytidine deaminase enzyme involved in somatic hypermu-361 tation and class switch recombination, which, in turn, results reduced.” The sentence looks incomplete.

6.       Table 1: Convert all “+” sign to superscripts.

7.       Section 4: The authors could consider including imiquimod, which when used as a topical cream has been shown to boost immune response of dermal influenza vaccination by mediating the TLR pathway.

8.       A table or figure summarizing approaches to reverse immunoaging would be perfect. After all, this review aims at discussing how to improve vaccination response (in older adults) as the title suggests.
